# Consumers’ Understanding of Ultra-Processed Foods

**DOI:** 10.3390/foods11091359

**Published:** 2022-05-07

**Authors:** Juliana Sarmiento-Santos, Melissa B. N. Souza, Lydia S. Araujo, Juliana M. V. Pion, Rosemary A. Carvalho, Fernanda M. Vanin

**Affiliations:** 1Laboratory of Bread and Dough Process (LAPROPAMA), Food Engineering Department, Faculty of Animal Science and Food Engineering (USP/FZEA), University of São Paulo, Av. Duque de Caxias Norte 225, Pirassununga 13635-900, SP, Brazil; julianasarmiento@usp.br (J.S.-S.); nascimentomelissabs@usp.br (M.B.N.S.); lydia.araujo@usp.br (L.S.A.); 2NOZ Pesquisa e Inteligência, Rua Cauowaá, 1575-92, São Paulo 01258-011, SP, Brazil; juvanin@nozinteligencia.com.br; 3Food Engineering Department, Macromolecules Functionality Multi-User Center (CEMFUM), Faculty of Animal Science and Food Engineering (USP/FZEA), University of São Paulo, Av. Duque de Caxias Norte 225, Pirassununga 13635-900, SP, Brazil; rosecarvalho@usp.br

**Keywords:** offer intention, NOVA classification, baby food, consumer knowledge, food processing level

## Abstract

Food classification systems have been proposed to improve food quality criteria. Among these systems, “processing level” has been used as a criterion. NOVA classification, as the denotation “ultra-processed” food (UPF), has been widely used in different countries. However, even though some studies have pointed out some controversial aspects, no study has evaluated its comprehension by the population where it is used as reference. Therefore, this study explored the understanding of the term UPF for Brazilian consumers, where this denotation has been used in the last 8 years. A questionnaire was used, with questions referring to different aspects of self-assessment of knowledge about UPF. Altogether, 939 valid participants completed the questionnaire, and 81.9% of them declared to know the term UPF. For 78.2%, a better definition for UPF should be “foods that have gone through many processes in industry”. Finally, it was concluded that the term UPF is still confusing for most Brazilians, indicating the risk of use and the urgent necessity to improve the classifications systems and consequently consumer understanding. Only when all parties interested in healthy food work together could this problem be solved.

## 1. Introduction

Changes in consumer diets have been observed in recent years. In general, there is an increase in the consumption of processed foods, a reduction in the intake of fresh products and an increase in the number of people who eat outside the home [1,2,3]. The diversity and increased supply of processed foods can influence the population’s eating patterns, especially children, as the first years of life stand out as an especially important period for establishing eating habits [4,5,6,7]. Parents identified time as a key difficulty preparing home-cooked meals with fresh ingredients [6]. Sadler et al. [8] consider that the increase in the number of working women, together with the need to save time, may be related to increasing development in the processed food market. Accordingly, it can be said that processed foods have been preferred due to their characteristics such as practicality and convenience. However, the healthiness of these products has been increasingly questioned.

Food classification arrangements that categorize foods considering their “processing level” have resorted to predict diet-related quality criteria and their outcome on consumer health.

Particularly, the NOVA classification presented the denotation “ultra-processed” food (UPF), and later, the dietary guidelines of many countries have used this classification system, including in Latin America, such as Brazil [9], Ecuador [10], Peru [11] and Uruguay [12], as well as Belgium [13] and France [14], in addition to the Food and Agriculture Organization of United Nations [15].

Basically, the NOVA classification groups foods into four major groups: (1) in natura or minimally processed foods, (2) processed culinary ingredients, (3) processed and (4) ultra-processed [16]. As claimed by NOVA, UPF are those that have five or more ingredients in their composition, mainly those with names unknown to the consumer, which are subjected to industrial processing that cannot be reproduced at home and that do not have an appearance or texture of a traditional food [9,16]. However, according to the authors themselves, this system does not consider the nutritional aspects of foods nor the techniques applied in the preparation of foods.

Some studies have demonstrated the risks and concerns that the application of these definitions can bring to consumer health [17,18], in addition to highlighting inconsistencies and flaws, since their definitions are broad and ambiguous and often not supported by scientific evidence [19,20]. Jones [18] considered that consumers are confused by the definitions put forward by NOVA and, for example, they are avoiding some foods considered as UPF, such as cereals and wholegrain/fortified breads, which can result in reduced intake of folate, calcium and fiber. Similarly, even though breastfeeding is considered as ideal, secure choices are indispensable when this is not possible. Jones [18] mentions that avoiding the consumption of infant formulas, food supplements and gluten- or lactose-free foods, as they are UPFs, may not contribute to improved health, in addition to being foods that are not considered in preventing the increase in obesity.

Furthermore, confirmation of the relationship between processed foods and healthiness is imprecise. It appears that classification systems were established to produce suggestions about larger groups of processed foods to draw generalized conclusions about the industrialized process. According to Poti et al. [21], although there is “very consistent” indication that UPF consumption and the levels of obesity and cardiometabolic health, it remains unclear whether this is process-dependent and not related to the nutritional content. The authors themselves conclude that more longitudinal studies are needed to control this confusion [21].

As previously mentioned, the Food Guide for the Brazilian Population (FGBP) was established considering the NOVA classification, and consequently, the guidelines use the theory of UPF. However, to be successful, healthcare statements must deliver easily comprehensible messages in order to be used by consumers and construct efficient assessments [22].

Thus, despite the considerable increase in studies on the NOVA classification, according to the authors’ knowledge, no study has been published in the literature exploring the understanding of the classification and the term “ultra-processed” in the Brazilian population. Therefore, this study aimed to explore how consumers conceptualize the UPF definition and assess the level of agreement between foods considered as UPF and those included in the NOVA classification. Based on the results obtained by the present survey, it is expected to provide subsidies to improve the knowledge in the light of the demystification of issues that have been gaining increasing social visibility, which in turn could impute better communication with the general population and all parties interested in food aspects, such as food industry, consumers and nutritional and health professionals.

## 2. Materials and Methods

### 2.1. Research Participants

The understanding of the term UPF was explored through the dissemination of an online questionnaire. The questionnaire was previously evaluated by the Research Ethics Committee of the University of São Paulo for further data collection (Certificate of Ethical Appreciation number 36600620.9.0000.5422). Subsequently, the questionnaire was disseminated via email and social networks between October 2020 and July 2021, using convenience sampling (snowball technique), a method that has become popular for recruiting participants by taking advantage of social media, by assuming that a qualified participant shares an invite with other participants. This process assumes that there is a link between the initial sample and the others, allowing a series of references to be made within a circle of acquaintances [23,24].

In total, 1195 participants answered the questionnaire, representing a fixed percentage (0.0005%) of the total population in Brazil [25]. Table 1 presents the socioeconomic and other characteristics of the participants. No information about the study objectives was provided to the participants. After detailed analysis of all responses, eliminating the incomplete ones, 939 valid responses were considered, and the final sample was non-stratified. Appendix A details the valid responses, preserving the participant’s privacy according to the Free and Informed Consent Term (TCLE).

### 2.2. Data Acquisition

The questionnaire was divided into 6 steps: (1) affirmation of self-knowledge about UPF; (2) identification of UPF with images; (3) confirmation of knowledge about UPF based on the guidelines of the Food Guide for Brazilian Population (FGBP) [9]; (4) intention to offer some foods to children; (5) presentation of the definition of UPF according to the FGBP and verification of agreement or not; and (6) socioeconomic data.

In the first step, the participants showed their self-knowledge about the subject by answering two questions: (1) “Do you know what UPF is?”, being able to answer “Yes” or “No”, and later indicating (2) “What do you think is the best definition of UPF?”, with two possible answers (a) “Foods with many processes performed by the food industry” or (b) “Foods with many ingredients in their formulation”.

The second step consisted of visual assessment, through images, of different foods. Participants answered if they consider the samples as UPFs or not. Six images were used: potato chips with salt; baby food—identified as industrialized organic; infant formula; apple—presented in packaged fruit format; soft drink; and loaf bread.

Four affirmations were presented, in the third step, to assess the participant’s knowledge about the concept of UPFs, and the participants questioned if they consider the information true or false. Then, the intention to offer certain foods to children under two years of age was evaluated (fourth step).

Finally, in the fifth step, the participant was presented to the definition of UPF as detailed in the FGBP [9]. Then, participants answered if they already knew these definitions, if they agree with it or do not agree and, finally, explained why they agree or not with the definition. In the sixth step, the participants provided data with multiple-choice questions aimed at ratings on socioeconomic issues such as gender, year of birth, education, income and place of residence. Participants were also asked about the presence of children in the family. The study included supplementary questions related to food habits; however, they were not analyzed in the present study.

### 2.3. Data Analysis

Results were analyzed by four researchers independently, and a consensus was reached for subjective differences.

In data analysis, Spearman correlation rank coefficient was initially used to evaluate the correlation of the knowledge of the participant with their characteristics (normality of data sets was assessed using the Anderson–Darling test with *p*-values below 0.005). Then, Pearson’s correlation was used to evaluate the correlation between the declaration of self-knowledge and the correct answers, as the normality of both data sets was assessed using the Anderson–Darling test with *p*-values of 0.05 and 0.95, respectively. In relation to the number of correct answers and the participants’ professions, the normality of both data with the same test had *p*-values below 0.005; therefore, as the data set does not follow a normal distribution, Spearman correlation rank coefficient was used [26]. Data were verified with Minitab software version 17.1.0 (Minitab, LLC; State College, PA, USA).

## 3. Results

The data presented in this research have a margin of error of 2.8%, with 95% confidence. In this study, the participants were sampled in a convenience sampling method, and 80% of participants were women. Therefore, the sample does not represent the Brazilian population. It could be suggested that the participants may reflect part of Brazilian consumers, especially women, and the results obtained may guide future studies with a representative sample of the Brazilian population. The higher number of responses from female participants (80.6%), as well as with children (63.3%), may be related to concerns about health and food, resulting in a high interest in the questionnaire. Furthermore, in relation to socioeconomic distribution, most of the participants, 87.4%, reside in the Southeast region, with an income between 1 and 3 (32.5%) or 4 and 6 (21.1%) minimum wages. These aspects, residence region and income, underlined that the results of the survey are most representative for the portion of the population in the region with the highest purchasing power and technology development in Brazil.

In relation to the highest averages of correct answers, it could be observed that women participants and residents of the Northern regions were those who had the highest number of correct answers (Table 1). The higher education of the participants and the presence of children in the family showed a very weak correlation with the increase in correct answers in the questionnaire. However, none of the characteristics of participants presented a strong correlation with the knowledge about UPF, through Spearman correlation. As all *p*-values were greater than the significance level of 0.05, no correlation was statistically significant could be proposed.

### 3.1. Participants’ Understanding Regarding the Term UPF

Most of the participants, 81.9%, declared to know the term UPF. Regardless of declaring whether they knew the term UPF or not, 78.2% of participants indicated that the best definition for UPF should be foods that have many processes performed on them by the food industry (Figure 1).

### 3.2. Checking Food Classified as UPF

Figure 2 shows the distribution of participants’ answers regarding the classification as UPF or non-UPF. Among the different foods presented, soft drinks and packaged fruits showed a clear convergence of opinion for UPF (92.3%) and non-UPF (94.8%), respectively, as the NOVA classification proposal. The participants also classified infant formula as UPF (80%).

For the other foods presented, such as baby food, loaf bread and potato chips with salt, it was observed that, in general, consumers are confused about the UPF classification, as 65.6%, 55.6% and 62.3% of the participants, respectively, classified them as UPFs (Figure 2). Thus, no clear evidence of classification was verified, as the numbers are close to half of the participants, not indicating a clear trend. According to NOVA classification, industrialized organic baby food, loaf bread and potato chips with salt are classified as UPF, UPF and processed, respectively. In the FGBP, potato chips with salt are also classified as processed [9]. Packaged fruits were classified as UPF by 5.2% of the participants (Figure 2).

Table 2 presents the distributions of participants’ answers related to the affirmations presented in the questionnaire. Among the four affirmations presented (Table 2), only affirmations 2 and 4 had a significant percentage of correct answers (87.7% and 63.9%, respectively), showing that the consumers’ perceptions about UPFs are related to products with many ingredients in their formulation, with high levels of sugar, fat, salt and additives, in addition to low nutritional quality.

On the other hand, the affirmations that presented examples of foods (1 and 3) showed significant errors, reinforcing the difficulty related to the classification of food as proposed by NOVA (Table 2). Yogurts and cereal bars are foods cited as examples of UPF by NOVA classification [9,16], and only 55.6% of the participants got the statement right.

In relation to the affirmation “Foods pre-processed by the industry with frying or cooking, such as frozen potatoes and broccoli, for example, are UPF” (Table 2), 73.9% of the participants considered this affirmation as true. However, according to NOVA classification, those products are UPFs and minimally processed foods, respectively.

### 3.3. Knowledge and Perception of Participants about UPF

Finally, the relationship between the participants who answered knowing the term UPF and the number of correct answers were, therefore, correlated by Pearson’s correlation, obtaining a medium level of correlation, 0.533 (*p*-value = 0.092) (Table 3). These results may indicate that even the participants believe to know UPF, in fact they do not have real knowledge about the subject.

The classification difficulty was also reported by the participants of the present questionnaire when they were asked to explain if they agree or not with the definition of UPF. Among the participants who answered 80% or more of the questions presented in the questionnaire correctly, that is, who demonstrated that they actually know what UPF is, some statements requesting a more understandable definition mention that “to reach the greatest number of people, so that it is understandable and more easily absorbed, such a definition lacks clarity” and also “should be written in a more accessible/understandable way for the general population”. There were also participants who mentioned that “they are foods with low nutritional quality” and that the definition “is not related to the amount of processes involved in food preparation”, in addition to “uses the term processed in the name but the definition refers to others characteristics” and “is not related to the processing but to the composition of the food” with information such as “the quantity of ingredients is not related to the processing of the product” and also that “a food could have many ingredients and not be UPF”. Many participants mentioned not understanding the subject clearly, writing that “it is not clear” even after reading the definition.

### 3.4. Intention to Offer Food for Children

When the participants were asked about the intention to offer certain foods to children, they chose only foods that were previously correctly classified (fruits and soft drinks), presenting results according to FGBC [27] (Figure 3). In other words, 91.9% of participants said they never offer a soft drink to a child under 2 years old, and 93.6% said they offer fruit without restriction. Even so, it is alarming to find that 1.0% of the participants answered that they never offer a fruit to a child under 2 years old; after all, fruits and vegetables are the basis of infant feeding, especially in the introduction of food [27,28,29,30].

A considerable number of participants considered never offering strawberry yogurt (36.5%), corn starch biscuits (33.6%) and tapioca flour biscuits (25.6%) to a child under 2 years old, which are classified as UPFs according to NOVA classification. This thought may be related to guidelines received by pediatricians.

In relation to infant formula and industrialized baby food, most of the participants considered offering those products “sometimes”, 84.7% and 60.1%, respectively. The results showed that most of the participants that classified infant formula (80.7%) and baby food (53.8%) as UPF (Figure 2) [9], they also intended to offer these foods to children. Such acceptance may be related to the consumer’s perception that the nutritional contribution of food is important, and not how this food is processed [18]. On this subject, the participants declared that “I would not stop giving my son the (infant) formula because it was UPF”, “the (infant) formula, I had to offer because he has APLV (cow’s milk protein allergy)”, and “I think the issue of recommendation is a bit delicate, that infant formula should be avoided”. There were also manifestations questioning the different classification when “not considering UHT (ultra-high temperature) milk as UPF and considering infant formula”. Furthermore, an important point was raised when considering that “the fact of not having exclusive breastfeeding can already be considered a stressful factor for a puerperal woman and she feels bad knowing that she is feeding her child with UPF, I think it’s unnecessary”.

### 3.5. Relation between UPF Understanding and the Participants Qualification

Considering the 10 questions presented related to the NOVA classification, the average of the correct answers for all valid participants was 6.2. The highest number of correct answers was observed among Gastronomy and Nutrition professionals. However, the correlation between the number of correct answers and the participants’ professions, through Spearman’s correlation, showed a very weak correlation value, 0.072 (*p*-value = 0.017). Therefore, regardless of the academic background, the participants’ knowledge about UPF is unsatisfactory, even for food or health professionals. Furthermore, even after reading the definition of ultra-processed products, the participants in the present research mentioned that it was confusing to understand and that the language is not “accessible to all audiences”.

## 4. Discussion

The definitions of UPF found in the literature show variability [18,19] and can open different interpretations [19], in addition to demonstrating a lack of agreement between the studies related to NOVA [18]. The present study gives substantial contribution in the research area on the lack of understanding about the NOVA classification, especially in relation to UPF. It is important to underline that it was expected to find higher knowledge about the term UPF, since most participants had at least completed higher education (78.4%). It is also important to highlight that most of the participants reside in the richest region of Brazil (Southeast). This aspect could be related to the use of the snowball technique and the fact that the questionnaire was launched by residents of the Southeast [24].

A similar result, as reported in Section 3.1 (Figure 1), was found with consumers in Uruguay, where 91.2% of the participants declared to know the term UPF and described it as highly processed products [31]. In addition to Uruguay, consumers in Argentina (*n* = 120) and Ecuador (*n* = 61) also related the term UPF to processing (92.0%), and not to the list of ingredients (21.2%) [32]. According to Sadler [8], divergences about the “degree of processing” probably stem from singular perceptions and intents once most of the classifications were proposed by epidemiologists.

Derbyshire [17] analyzed 50 foods that meet the definition of UPF by NOVA and found that they are classified as healthy according to the parameters of the Nutrient Profile Model in the United Kingdom. Furthermore, for the same foods, there was no correlation between the number of ingredients and nutritional quality when compared to the nutritional profile of European Regulation No. 1924/2006 [33].

As detailed in Section 3.2, some products are very difficult to be classified in the NOVA classification. Ares et al. [31] reported that breads, French fries, fried foods and packaged foods, in general, were not clearly classified and may depend on specific product characteristics such as ingredients and nutritional composition. Aguirre et al. [32] also considered the classification of bread as non-obvious, and only 3.9% of the participants considered bread as UPF. This difficulty was also reported for bread by nutrition students and professionals, and only 11% of them classified it correctly [34]. In the case of packaged foods, Ares et al. [31] showed that 4.0% of participants considered packaged foods as UPF, corroborating with this study. Similarly, frozen fruits and vegetables were also considered highly processed by low-income parents [35].

The disagreement about frozen vegetables underlined again the food classification difficulty by consumers when trying to use the NOVA classification. Furthermore, the definition of UPF has been updated over the years [36,37]. The FGBP published in 2014 explained that “the processing techniques used in the manufacture of UPF include pre-processing with frying or cooking” [9]. Thus, it can be understood that pre-processing with bleaching, in the case of frozen broccoli, results in considering this food as a UPF. Later, publications indicated that only pre-processing with frying must classify food as UPF [38]. The classification of frozen foods was also considered unclear by Ares et al. [31].

Another product that deserves special attention is infant formulas. There is no doubt that breast milk is the best option [9,39]; however, experts believe that the second-best option is infant formula [39]. In Brazil, the recommendation of the Food Guide for Children Under 2 years old, published in 2019, to offer cow’s milk when breastfeeding is not possible due to the classification of infant formulas as UPF [9,27] was questioned by the Brazilian Society of Pediatrics. Finally, in 2021 an abbreviated and revised version of the Food Guide for Children Under 2 years old was published, indicating that infant formula in Brazil can be used when necessary [40]. In addition to not recommending cow’s milk to infants before 12 months of age, due to difficulty in digestion, infant formulas are recommended as the safest option in the United States [39] and in Europe [41] when breastfeeding is not possible. It is also important to highlight the danger in the search for homemade infant formulas to replace industrialized infant formulas [18,41] by saying that homemade preparations are healthier than industrial ones. In this regard, the Food and Drug Administration warns of profoundly serious problems when preparing and offering homemade infant formulas to babies, the consequences of which range from severe nutritional imbalances to foodborne illnesses, which can be fatal [42].

Unfortunately, baby food was intended to be offered without restriction for only 11.1% of the participants, less than strawberry yogurt (13.0%), corn starch (12.7%) and tapioca flour biscuits (16.7%). Most industrialized baby foods sold in Brazil do not contain salt in their composition, only vegetables, meats, legumes, cereals and fruits, among others [43,44,45]. The FGBC recommends that children’s meals should have one food from each group: (1) beans; (2) cereals or roots or tubers; (3) meat or eggs; (4) legumes and (5) vegetables [29]. Following the recommendation of the FGBC [27], if an industrialized baby food has rice, beans, Basella alba, pumpkin and egg in its formulation, sauteed with oil, onion and garlic, it is already classified as UPF because it has more than five ingredients, even without any additives such as dyes, preservatives, antioxidants, flavorings, flavor enhancers and sweeteners, among others.

Commercial baby food appears to be equally safe and healthy as homemade baby food [46]. Brembeck and Fuentes [47] studied 19 mothers in Sweden and concluded that there is no contradiction between care and convenience, as good motherhood is no longer unilaterally associated with home cooking. These mothers reported that convenience is when the husband takes on the responsibility of cooking, having good baby food on hand, having a child who feeds himself using cutlery or having the whole family eat the same food at the same time [47]. For mothers, convenience food is not necessarily processed food. In addition, there are few types of food available in the Brazilian market for children in early childhood. The supply of foods not suitable for age is a concern [48,49] and not necessarily the quality of baby food, as these foods follow strict regulations.

The suspicion mentioned in Section 3.4 about guidelines received by pediatricians was also observed in a survey carried out in Uruguay that investigated whether the 212 pediatricians’ recommendations to parents during the introduction of food were in accordance with current guidelines. This study showed that UPFs considered less obvious and judged to be frequently consumed by children, such as yogurt, dairy desserts and biscuits, were not mentioned by pediatricians when listing foods that must be avoided [50]. Additionally, a survey conducted in Italy with 509 children and adolescents showed that sweet biscuits (13.2%) and processed beverages (9.3%) are among the main UPFs consumed [51]. It is also worrying that 34.3% of the participants considered offering processed juice to a child under 2 years old, knowing that processed juices are among the products with the highest levels of added sugar [52].

Following the difficulties and confusing about NOVA classification, the intention to correlate knowledge about UPF with the participants’ professional training (Section 3.5) has already been addressed by recent research showing the lack of credibility of data generated by researchers and the possibility of incorrect guidance to the population. Studies carried out with students and nutrition professionals in a Brazilian higher education institution in 2015 (*n* = 72) [53] and 2016 (*n* = 69) [34] showed that these students and professionals had unsatisfactory knowledge about the NOVA classification, concluding as a critical point, as it should be easily understood by everyone, regardless of their background. Similar results were found in France [54], with 150 specialists in human nutrition and/or food technology. In this regard, NOVA classification has been questioned by several studies due to its complexity [30,32], inconsistencies [54,55] and broad and ambiguous definitions [8,17,19]. Although classification systems have been used and considered in public health policies, this study explored that its implementation by national or international organizations does not, therefore, mean that no additional studies or criticism is needed. In general, it is observed that there is no clear consensus in classifications about which characteristics make a food more or less processed.

From the nutritional point of view, some studies have reported that the higher levels of sugar intake in the diet are associated with homemade products such as tea and coffee [52]. Additionally, Derbyshire [17] showed that many foods that are classified as UPF by NOVA are not considered “less healthy” according to the United Kingdom nutrient profile model. These types of approaches allow reflecting on targeted public health strategies, suitable for different groups of people. All of this underlined that the use of claims such as “homemade”, among other marketing tools, has been growing in products that could be less healthful than industrialized ones [56].

From the point of view of technology and food science, the industrial process must always be improved, aiming to maximize the positive effects and minimize the negative ones, which in turn can be much better measured and controlled in the food industry than at the domestic level. Focusing only on the promotion of cuisine negates the need for the food safety, food security, convenience and practicality that is necessary for the evolution of contemporary social systems. Re-analyzing our understanding about processed foods and their relationship with the social needs of contemporary life represents a path to be built together by food engineers, nutritionists and epidemiologists in order to avoid biased traps that indicate a faster but more dangerous path.

## 5. Conclusions

For the first time, the comprehension about UPF definition was evaluated with the Brazilian population, although it has been used in Brazil over the last 8 years in the Food Guide for the Brazilian Population. The term UPF is still confusing for most participants. Furthermore, considering the socioeconomic characteristic of participants, it could be concluded that the definition of UPF is not clear for those who had at least completed higher education or reside in regions with higher levels of development.

This confusion seems to be related to the definitions associated with the term, as this is not logical and therefore not easily understandable and may induce the consumer to make mistakes when making a purchase decision.

Consumers are avoiding some foods considered as UPF, even products which could contribute to improved health, which in turn could be related to the increase in some food phobia. All of this underlined the risk of consuming UPF to consumer health. Healthier choices could ensue with the development of new products, however, the manner in which processed foods have been classified does not motivate innovation in the food industry. A food classified as UPF, even if reformulated for health aspects, would still be classified as UPF simply because it was industrially produced.

Care is needed to balance naïve and heuristic messages with scientific rigor and avoid unwanted consequences. All parties interested in adequate food should improve the classification system and, consequently, the understanding of the consumer; after all, innovation with healthy, sustainable, safe and convenient foods could greatly benefit the population.

## Figures and Tables

**Figure 1 foods-11-01359-f001:**
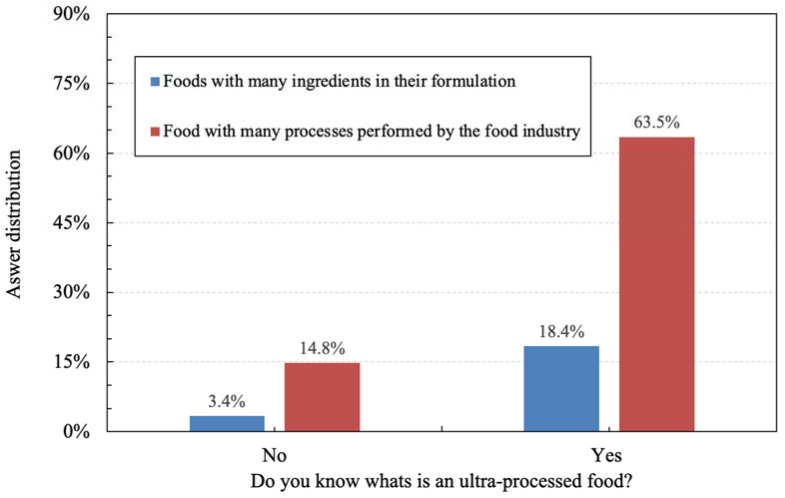
Distribution of answers from valid participants (*n* = 939) to the question “Do you know what is an ultra-processed food?”, with further indication of the best definition for ultra-processed food (UPF) by the participants.

**Figure 2 foods-11-01359-f002:**
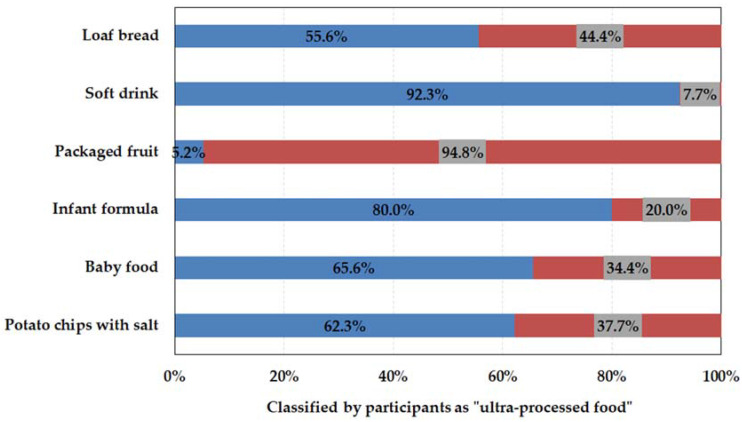
Distribution of food classification presented to research participants as UPFs (blue) or not (red).

**Figure 3 foods-11-01359-f003:**
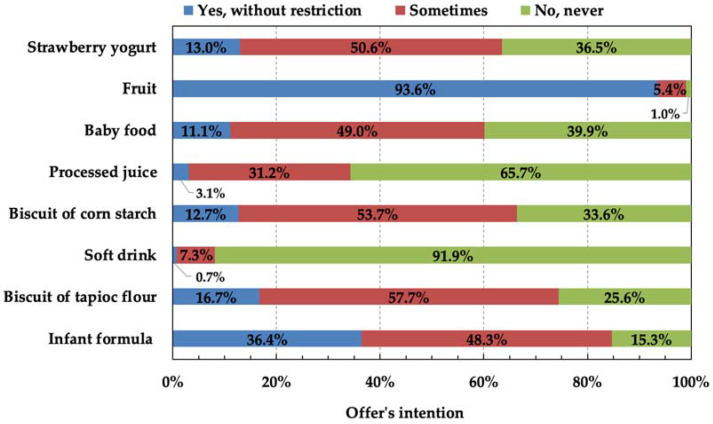
Distribution of intention to offer food to children under two years old.

**Table 1 foods-11-01359-t001:** Socioeconomic and characterization of participants.

Characteristics	Absolute Frequency	Relative Frequency	Average Correct Answers	Spearman Correlation	*p*-Value
Gender				−0.072	0.031
Female	756	80.6%	6.6		
Male	181	19.2%	6.4		
Other	2	0.2%	4.0		
Age				0.000	0.994
Under 18	0	0.0%	-		
18 to 24	178	19.0%	6.5		
25 to 34	367	39.1%	6.6		
35 to 44	282	30.0%	6.6		
45 to 54	76	8.1%	6.1		
55 to 64	27	2.9%	7.1		
>65	9	1.0%	6.1		
Children in the family				0.018	0.583
No child	345	36.7%	6.6		
With child (<17 years old)	594	63.3%	6.8		
Education				0.011	0.745
Complete/incomplete elementary school	40	4.3%	6.5		
Complete high school	163	17.3%	6.5		
Complete higher education	736	78.4%	6.7		
Income				−0.076	0.021
Without own income (now)	112	11.9%	6.6		
Below minimum wage	34	3.6%	6.2		
From 1 to 3 minimum wages	305	32.5%	6.8		
From 4 to 6 minimum wages	198	21.1%	6.6		
From 7 to 9 minimum wages	109	11.6%	6.5		
Above 10 minimum wages	139	14.8%	6.3		
I prefer not to inform	42	4.5%	6.5		
Region of residence				−0.042	0.208
North	27	2.8%	7.8		
Northeast	20	2.2%	6.2		
Southeast	820	87.4%	6.5		
Midwest	14	1.5%	6.5		
South	58	6.1%	6.1		

**Table 2 foods-11-01359-t002:** Percentage of correct answers and errors related to the affirmations presented in the questionnaire.

Affirmation	True (T) or False (F)	Correct	Error
Examples of UPF are: yogurts and cereal bars. (1)	T	55.6%	44.4%
UPFs are nutritionally rich and low in calories, sugar, fat, salt and chemical additives, with enhanced flavor and longer shelf life. (2)	F	87.7%	12.3%
Food pre-processed by the industry with frying or cooking, such as frozen potatoes and broccoli, for example, are UPFs. (3)	F	26.1%	73.9%
Products with industrial formulations consisting of five or more ingredients are UPFs. (4)	T	63.9%	36.1%

**Table 3 foods-11-01359-t003:** Relation between participants who answered knowing or not ultra-processed foods (UPFs) and the number of correct answers presented (identification of UPFs by images and statements to identify true or false).

Do You Know What Is an UPF?		Number of Correct Answers in the Questionnaire	Total
Yes	No	(Images and Affirmations)	
1	0	1	1
5	0	2	5
23	2	3	25
41	21	4	62
114	30	5	144
159	36	6	195
174	31	7	205
170	35	8	205
73	12	9	85
9	3	10	11

## Data Availability

Data is contained within the article and Appendix A.

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
