# Peer review of "Consumers’ Understanding of Ultra-Processed Foods"

_foods, 2022, doi:10.3390/foods11091359_

Round 1

Reviewer 1 Report

The objective of this study was to classification of food to improve food quality criteria. Among these systems, “processing level” has been used as a criterion. NOVA classification, as the denotation “ultra-processed” food (UPF), has been widely used in different countries.

Thematically the work is interesting for the researchers and professionals and the proposed manuscript is relevant to the scope of the journal.

Some modifications and clarification must be made by the Authors.

The overall organization and structure of the manuscript are appropriate. The paper is well written and the topic is appropriate for the journal.
The aim of the paper is well described and the discussion was well approached, its results and discussion are correlated to the cited literature data.

The literature review is comprehensive and properly done.

The novelty of the work must be more clearly demonstrated.

Statistical interpretation of the analytical data must be more properly presented.

The verification of the model should be performed. 

Other Specific Comments: The work is properly presented in terms of the language. The work presented here is very interesting and well done, it is presented in a compact manner.

The main drawback of the paper i s the extent of novelty, or the main novelty in the present work, compared to the works of other researchers? In my opinion, the authors should put additional effort to demonstrate that the present work gives a substantial contribution in the research area.

The manuscript should be improved from technical/graphical viewpoint.

Author Response

Dear Reviewer 1,

Thank you very much for assessing our manuscript and for the useful comments and suggestions on the manuscript. We modified the manuscript accordingly to your points and changes in the manuscript were highlighted. In addition, answers to your comments are in the attached file.

Reviewer 2 Report

This study aimed to explore the understanding of the term UPF of Brazilian consumers. The aim seems interesting but there are some issues that should be addressed. Here are comments especially regarding method and results.

- Study subjects

  • It is hard to understand what the authors are trying to say in lines 101–103.
  • It would be better to move the content described in lines 142–149 into the method section.
  • If 939 participants are the samples to be analyzed, the sample size explained in the abstract should be changed to 939, not 1195.
  • In this study, the participants were sampled in a convenience sampling method. Eight in ten participants were women. Moreover, the target population was not the Brazilian population. Thus, it seems unreasonable to mention about the generalizability of the results (lines 150–153).

Method and results

  • There was no mention on collection of socioeconomic data.
  • Please delete the instruction for authors (lines 139–141).
  • Was there any special reason for four researchers to independently perform analyses? (lines 129–130) If so, please explain the reasons. This type of analysis seems uncommon.
  • Please, present the frequency (no) with % in Table 1.
  • It is recommended that you move the interpretation of the results (eg, lines 156–161, 204–213, 249–259, 263–274) to the Discussion section.
  • It is hard to understand what the authors are trying to show in Figure 1. Is this result for all study participants (n=939) or for those who were aware of the term UPF?
  • Please check the results. For now, the numbers between the text and the figure are different. (eg, Please see lines 181–188 and figure 2; line 235–249, 260–262, and figure 3.)
  • In lines 276–287, from what I understood, the authors tried to show whether the UPF understanding were different by the participants’ jobs. The variable of job is generally known as categorical. Correlation analysis does not appear to be appropriate for these data and an analysis purposes.

- Please, present the reference information in English.

Author Response

Dear Reviewer 2,

Thank you very much for assessing our manuscript and for the useful comments and suggestions on the manuscript. We modified the manuscript accordingly to your points and changes in the manuscript were highlighted. In addition, answers to your comments are in the attached file.
